# Empirical calibration of a simulation model of opioid use disorder

**R. W. M. A. Madushani[1], Jianing Wang[2], Michelle Weitz[1], Benjamin P. Linas[1,3], Laura F. White[2], Stavroula A. Chrysanthopoulou[4]***

1 Section of Infectious Diseases, Boston Medical Center, Boston, Massachusetts, United States of America, 2 Department of Biostatistics, Boston University School of Public Health, Boston, Massachusetts, United States of America, 3 Department of Medicine, Boston University School of Medicine, Boston, Massachusetts, United States of America, 4 Department of Biostatistics, Brown University School of Public Health, Providence, Rhode Island, United States of America

* stavroula_chrysanthopoulou@brown.edu

## Abstract

### Background

Simulation models of opioid use disorder (OUD) aim at evaluating the impact of different treatment strategies on population-level outcomes. Researching Effective Strategies to Prevent Opioid Death (RESPOND) is a dynamic population, state-transition model that simulates the Massachusetts OUD population synthesizing data from multiple sources. Structural complexity and scarcity of available data for opioid modeling pose a special challenge to model calibration. We propose an empirical calibration approach applicable to complex simulation models in general.

### Methods

We implement an empirical approach to calibrate RESPOND to multiple targets: annual fatal opioid-related overdoses, detox admissions, and OUD population sizes. The empirical calibration involves Latin hypercube sampling for searching a multidimensional parameter space comprising arrivals, overdose rates, treatment transition rates, and substance use state transition probabilities. The algorithm accepts proposed parameters when the respective model outputs lie within pre-determined target uncertainty ranges. This is an iterative process resulting in a set of parameter values for which the model closely fits all the calibration targets. We validated the model assessing its accuracy to projections important for shared decision-making of OUD outside the training data.

### Results

The empirical calibration resulted in a model that fits well both calibration and validation targets. The flexibility of the algorithm allowed us to explore structural and parameter uncertainty, reveal underlying relationships between model parameters and identify areas of model improvement for a more accurate representation of the OUD dynamics.

**Data availability statement:** All relevant data for this study are publicly available from the GitHub repository (https://github.com/SyndemicsLab/PLOS_One_RESPOND_Data).

**Funding:** This work was funded by the National Institute on Drug Abuse (R01DA046527) (P30DA040500) and the National Institute of General Medical Sciences (R35GM141821). The funders had no role in study design, data collection and analysis, decision to publish, or preparation of the manuscript.

**Competing interests:** The authors have declared that no competing interests exist.

## Discussion

The proposed empirical calibration approach is an efficient tool for approximating parameter distributions of complex models, especially under complete uncertainty. Empirically calibrated parameters can be used as a starting point for a more comprehensive calibration exercise, e.g., to inform priors of a Bayesian calibration. The calibrated RESPOND model can be used to improve shared decision-making for OUD.

## 1. Introduction

Opioid use disorder (OUD) is an epidemic in the U.S and opioid overdose is a growing public health emergency [1–4]. Policymakers and public health officials seek effective strategies to address OUD and to reduce opioid death. Buprenorphine-naloxone, naltrexone, and methadone are all FDA-approved medications for OUD (MOUDs) that decrease opioid use, mortality, and criminal activity [5]. These treatments are under-utilized, with fewer than 25% of people with OUD on treatment [6]. One of the key pillars of the U.S. response to the overdose crisis is increasing access to MOUD [7,8]. This requires identifying and working in new venues and among new populations not currently engaged with treatment. Understanding the relative value of different venues, the budget impact of expanding MOUD, and the cost-effectiveness of various models of care are essential but unknown information for decision making.

Simulation models provide a tool for evaluating the impact of different care delivery and treatment strategies on population-level outcomes, integrating data from multiple sources to translate outcomes from clinical studies to policy-relevant information on population health and cost [9,10]. Models simulating state-level population estimates of people with OUD have been used to investigate delivery system innovations and project the impact on public health outcomes and cost, providing valuable insight into the fight against OUD [11]. However, due to the complicated nature of involved health systems, these models inevitably have a complex structure to capture real-world dynamics.

In previous studies, simulation models have been used to investigate the health and economic effects of prevention, treatment, or harm reduction interventions targeting opioid misuse and/or overdose [7,8]. For example, system dynamic simulation models have been used to project the effects of interventions to lower prescription opioid misuse on opioid overdose deaths, and to study policy implications of reducing the prescription dosage, increasing addiction treatments and lay naloxone use, etc., in the U.S [12,13]. An agent-based simulation has also been used to estimate the impact of evidence-based strategies for preventing opioid overdose deaths in a selected number of states [14].

Complex simulation models have been widely used in the literature as a promising tool for evidence synthesis, making projections, and informing decision analysis in the area of Public Health [10,15,16]. However, despite their advantages and flexibility, one of the biggest challenges in the development of this type of models is calibration and validation to real data, mainly due to the multiple sources of uncertainty involved in the process, the complex structure, and the lack of available data to inform every aspect of the process which the model aims to capture [11,17].

We have developed the Researching Effective Strategies to Prevent Opioid Death (RESPOND) model, with the goal of informing state-level innovation for low-barrier access to MOUDs in Massachusetts (MA). RESPOND is a dynamic, state-transition model, aimed at simulating the OUD dynamics and evaluating the effect of different interventions on preventing opioid overdoses. The model involves a set of distinct states and parameters representing observed and latent characteristics of the OUD epidemic. One essential challenge to modeling OUD and the overdose

crisis is the need to estimate parameter values for behaviors and events that are impossible to observe and/or are difficult to measure with bias-related to stigma and under-reporting of risk. The structural complexity of RESPOND combined with the scarcity of available data for opioid modeling poses a challenge to calibrating and validating the model [18,19].

In this study we present details of an empirical approach to calibrate the RESPOND model, as an example applicable to complex simulation models in general. We outline the key challenges related to calibrating any simulation model, describe the main steps involved in an efficient calibration process, and present the details from the implementation of an empirical calibration approach which can serve as an intermediate, effective process to a more comprehensive fitting of a complex simulation model.

## 2. Methods

### 2.1. The RESPOND model

We developed and calibrated the RESPOND model, a dynamic, cohort-based, state-transition model that simulates the OUD population in Massachusetts. This model simulates transitions of a cohort between distinct Markov states representing key aspects of the OUD dynamics under different interventions [9,10]. Fig 1 provides an overview of the RESPOND model structure. The simulation code is available at https://github.com/SyndemicsLab/RESPONDv1. The model simulates the clinical progression of a cohort with OUD assuming five major health states: overdose, death, and three treatment status states. Within each treatment state, there is a core simulation of four distinct OUD states determined by injection status (active, non-active, and injection, non-injection).

**Core simulation of OUD.** The core simulation of OUD includes four mutually exclusive states by the status and type of opioid use: 1) active, non-injection (ANI), 2) non-active, after non-injection (NANI), 3) active, injection (AI), and 4) non-active, after injection (NAI) (Fig 1). Active-use states are characterized by risk of overdose and elevated healthcare utilization, with active injection use having a higher risk than active non-injection (oral) use. Non-active use states carry no risk of overdose. The model simulates a bidirectional movement between active and non-active use states to capture the relapsing and remitting nature of opioid use.

**Simulation of treatment status.** The core simulation of opioid use is embedded in the treatment status simulations (Fig 1). RESPOND models acute detoxification services (detox) and three medication-based treatments for OUD (methadone, buprenorphine, and naltrexone). MOUD states have three dynamics modeled: 1) treatment initiation effect whereby individuals initially move out of active use, 2) subsequent bidirectional movement between use states, and 3) reduction in overdose risk.

The model simulates loss to follow-up from MOUD treatment by moving the population disengaging from MOUD treatment to the "post-treatment" health state, during which there is a high risk of relapse to active use states and consequently a high risk of overdose.

**Mortality.** There are two distinct sources of mortality in RESPOND:

1. Overdose mortality. There is a conditional risk of death from overdose. Overdose rates depend on drug use status and treatment and are stratified by age and sex.

2. Non-overdose mortality. The model employs standardized mortality ratios (SMR) that are stratified by sex and drug use status with elevated mortality among drug users using age-sex stratified actuarial life tables of the U.S [20].

**Cohort Initiation.** The RESPOND model has a weekly cycle. Every week, a new population "arrives" to the no-treatment state, with independent age- and sex-stratified entry rates. These rates represent new OUD cases and emigration of OUD cases to Massachusetts.

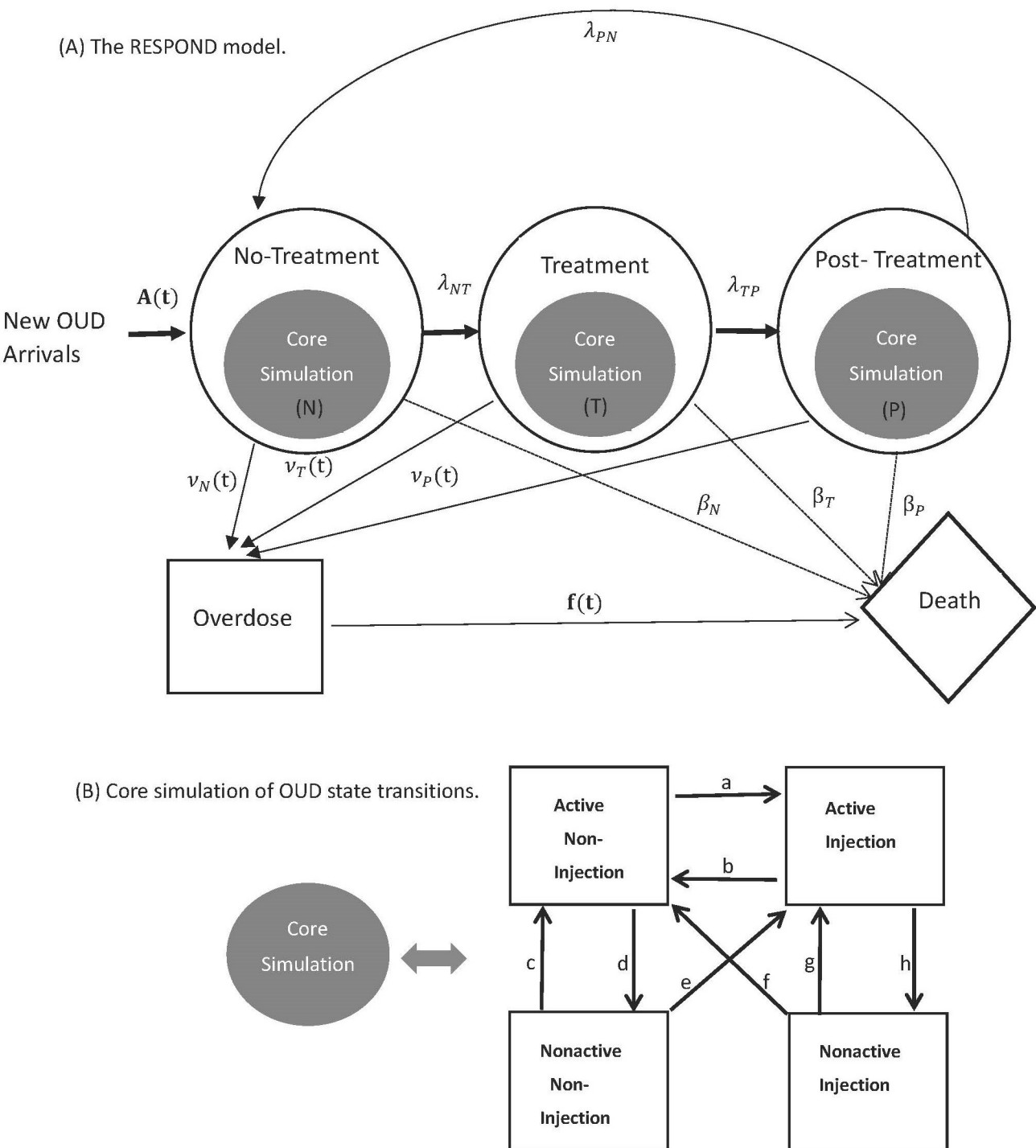

**Fig 1. Model Structure of the RESPOND Model. (A)** Overall model structure consists of three main health states: No-treatment, Treatment and Post-treatment denoted by $N, T$ and P respectively. Both T and P are stratified by four treatment types: buprenorphine, naltrexone, methadone, and detox. $A(t)$ denotes yearly time varying total new arrivals, and $\lambda$ denotes the transition rates between health states N, T and P. $\nu(t)$ denotes yearly time varying overdose rates parameter in health states, and $\mathbf{f(t)}$ denotes yearly time varying fatal overdose proportion applied on all overdoses (fatal and non-fatal combined) resulted from all health states. β denotes non-overdose related other cause mortality rates in health states N, T and P represented by dashed arrows. **(B)** Core simulation of substance use state transitions within each health state. Eight different transition probabilities in the core simulation are denoted by $a, b, c, d, e, f, g$ and h.

The only reason from exiting the simulation is mortality. The difference between these arrivals and deaths results in the simulated total Massachusetts OUD population size.

## 2.2. Data

The primary data source for RESPOND is the Massachusetts Public Health Data Warehouse (MA PHD). MA PHD is a longitudinally linked, administrative records database that includes service encounter and other administrative data from over 30 sources in Massachusetts. The individually linked database includes approximately 97% of the Massachusetts population. This database informed the calibration targets (Table 1) and several model input parameters (Table 2). We also used data from the National Survey on Drug Use and Health (NSDUH), and 2010 US Census to inform population counts of the initial cohort and new OUD arrivals demographic proportions. We used published studies on OUD natural history to estimate transitions between OUD no-treatment states. National Institute on Drug Abuse (NIDA) Clinical Trials Network (CTN) studies with weekly urine toxicology data inform treatment substance use transitions. We used data from the National Vital Statistics System, the 2010 US Census, and MA PHD to estimate non-overdose death rates. The RESPOND model is deemed non-human subjects research by the Boston University/Boston Medical Center Institutional Review Board (H-38867).

## 2.3. Model calibration

The model requires 52 different input parameters, some further stratified by age, sex, route of administration (injection vs non-injection), and/or time (Table 2, Fig 1). We calibrated parameters for which information (estimates from the literature or data sources) was either missing, unreliable, or highly uncertain. For example, since we used good quality data from NIDA CTN trials to estimate weekly substance use transition rates within MOUD treatments with an age and sex adjusted multistate Markov model, we did not calibrate those. However, Massachusetts state-specific no-treatment OUD state transition rates were not available, therefore we derived them from estimates provided in published studies on New York, California, and Maryland [24–26]. Similarly, due to lack of information, we assume the same transition rates as no-treatment for post-treatment except for transitions c (NANI → ANI) and g (NAI → AI), for which we assume higher rates post- compared to no-treatment. Transition rates c and g in all post-treatment types were derived from estimates published in a study of opioid detoxification patients [27]. Therefore, we calibrated OUD state transitions in no-treatment and post-treatments. We also had no

**Table 1. Calibration Targets.**

| Year | Total OUD Count*,& (95% CI) | Number of Fatal Overdoses** | Number of Admissions to Detox** |
|------|------------------------------|------------------------------|---------------------------------|
| 2013 | 226861 (222242, 231285)*** | 900 | 39635 |
| 2014 | 233184 (228680, 237514)** | 1294 | 41229 |
| 2015 | 275070 (272707, 277402) | 1562 | 38329 |

*Total OUD Count = Alive OUD + Fatal Overdoses + Other Deaths.

*"These are observed counts from MA PHD with no known uncertainty around the values. Therefore, we use within 10% of the values as the uncertainty interval for the empirical calibration

***Estimates of total OUD counts for years 2013 and 2014 were not used in the empirical calibration (i.e., for accepting/rejecting parameter values) due to missing an important data piece in MA PHD. Details about the estimates of Total OUD counts are in the Table.S1 in the supplemental material.

&Estimated from capture-recapture analysis [21].

**Table 2. Model parameters.**

| Parameter | Description | Data Source | Fixed vs Calibrated parameter (Reason) |
|---|---|---|---|
| $N_0$ | Number of people in no-treatment at the start of the simulation* | MA PHD, NSDUH and US Census 2010[&] | Fixed (High quality data) |
| $T_0$ | Number of people in treatments at the start of the simulation* | | |
| $P_0 = 0$ | Number of people in post-treatments at the start of the simulation* | Assumed | |
| $\alpha_{j,k}(t)$** | Age-sex stratified weekly new OUD arrivals: $\alpha_{j,k}(t) = \mathbf{A(t)} \times p_{j,k}(t)$ | Derived | |
| | A(t): weekly non-stratified total arrivals [b] | MA PHD, Barocas et al [21] [&] | Calibrated (Uncertain parameter with no direct estimates of uncertainty around derived point estimates and was found to be sensitive to model outcomes in preliminary analysis prior to calibration) |
| | $p_{j,k}(t)$: yearly time varying demographic proportions | NSDUH MRB Statistical Inference Report [22] | Fixed (High quality data) |
| $\lambda_{NT}(t)$** | Transition rate*** from no-treatment to treatments | MA PHD[&] | Fixed (High quality data) |
| | $\lambda_{NB}$: age-sex stratified rates from $N$ to buprenorphine | | |
| | $\lambda_{NX}$: age-sex stratified rates from $N$ to naltrexone | | |
| | $\lambda_{NM}$: age-sex stratified rates from $N$ to methadone | | |
| | $\lambda_{ND}(t) = \bar{\lambda}_{ND} \times \eta(t)$ where $\lambda_{ND}(t)$ denotes age-sex stratified rates from $N$ to detox | Derived | |
| | $\tilde{\lambda}_{ND}$: approximate transition rates from $N$ to detox (age-sex stratified) | MA PHD[&] | Fixed (High quality data) |
| | $\eta(t)$: time varying detox transition rate multipliers [b] | Range assumed (see Table 3) | Calibrated (Rates had to be increased to hit admissions to detox targets) |
| $\lambda_{TP}$ | Transition rate*** from treatment to post-treatment | Derived from estimates of published studies [6,23] [&] | Fixed (Well informed by reliable data) |
| $\lambda_{PN}$ | Transition rate*** from post-treatment to no-treatment calculated as $\lambda_{PN} = ln(1-0.25) \approx 0.29$. | Assumed | Fixed (Model assumes a constant rate for outflow from post-treatment) |
| $\gamma_T$ | Treatment Initiation effect [a]: the probability of keeping the current OUD state immediately after transitioning to a treatment (stratified by the route of administration: injection vs non-injection) | derived from CTN trial data[&] | Fixed (High quality data) |
| $\gamma_P$ | Post-treatment initiation effect [a, b]: probability of keeping the current OUD state immediately after transitioning to post-treatment (stratified by the route of administration: injection vs non-injection) | Range assumed (see Table 3) | Calibrated (No available data to derive post-treatment specific effects) |
| $\rho_N$ | OUD state transition probabilities within the core simulation of no-treatment [b] | Different published studies[&] | Calibrated (No available data to derive estimates for the state of Massachusetts) |
| $\rho_T$ | OUD state transition probabilities within the core simulation of treatments. Note: The model assumes everyone is non-active in detox | Estimated (except detox) from multi-state models using CTN trial data[&] | Fixed (High quality data) |
| $\rho_P$ | OUD state transition probabilities within the core simulation of post-treatments [b] | Derived from estimates of published studies[&] | Calibrated (No available data to derive post-treatment specific estimates for the state of Massachusetts) |
| $\nu_N(t)$ | Weekly overdose rates in no-treatment: $\nu_N(t) = o(t) \times m_N$ | Derived | |
| | $o(t)$: age-sex stratified weekly overall overdose (fatal and non-fatal) rate for the whole population at time t | MA PHD [&] | Fixed (High quality data) |
| | $m_N$: no-treatment overdose rate multiplier [b] and $m_N \in \mathbb{R}^+$ | Range assumed (see Table 3) | Calibrated (No available data to inform the rates of no-treatment) |

*(Continued)*

**Table 2.** (Continued)

| Parameter | Description | Data Source | Fixed vs Calibrated parameter (Reason) |
|---|---|---|---|
| $\nu_T(t)$ | Weekly overdose rates in treatments: $\nu_T(t) = \nu_N(t) \times m_T$ <br> Note: The model assumes no overdoses occur in detox | Derived | |
| | $m_T$: treatment overdose rate multiplier [b] and $m_T \in (0,1)$ | Range assumed (see Table 3) | Calibrated (No available data to inform the treatment specific rates) |
| $\nu_P(t)$ | Weekly overdose rates in post-treatments: $\nu_P(t) = \nu_N(t) \times m_P$ | Derived | |
| | $m_P$: post-treatment overdose rate multiplier [b] and $m_P > 1$ | Range assumed (see Table 3) | Calibrated (No available data to inform the post-treatment specific rates) |
| **f(t)**** | Weekly fatal overdose proportions [b] | MA PHD [&] | Calibrated (estimated from observed overdoses (both fatal and non-fatal) reported in MA PHD and uncertain due to under reporting) |
| $\beta_N$ | Non-overdose related other cause mortality rates in no-treatment estimated using standard mortality ratios (SMR)[&] | National Vital Statistics System, 2010 Census and MA PHD [&] | Fixed (not believed to have a significant impact on important model outcomes according to expert opinion) |
| $\beta_T$ | Non-overdose related cause mortality rates in treatments estimated using SMR[&] | | |
| $\beta_P$ | Non-overdose related cause mortality rates in post-treatment estimated using SMR[&] | | |

*Start of the simulation is beginning of year 2013.

**Time variability is in years not by weekly cycles (t = 2013, 2014, 2015).

***Note that RESPOND is a state transition model. Therefore, we convert all the rate parameters to probabilities using the formula $p = 1 - e^{-rt}$ which assumes a Poisson process with constant a rate for events.

[a]Substance use transition parameter within the core simulation. Upon transitioning to a health state there is an initiation effect specific to the health state which is then followed by transitions between OUD states within the core simulation.

[b]A calibration parameter.

[&]See RESPOND technical appendix

Notations: $j$ = age stratum; $k$ = sex stratum; $N$ = no-treatment; $T$ = treatment (buprenorphine, naltrexone, methadone, or detox); $P$ = post-treatment (post-buprenorphine, post-naltrexone, post-methadone, or post-detox)

information about annual new OUD arrival counts, while parameter estimates of overdose rates and fatal overdose proportions were derived from underreported overdose counts observed in MA PHD. Table 2 describes all the RESPOND model parameters summarizing the rationale for calibrating (or not). The calibrated parameters are the total number of new OUD arrivals in a week, multipliers on transition rates from no-treatment to detox, multipliers on overdose rates (fatal and non-fatal combined) in all health states, fatal overdose proportions, and age-stratified OUD transition probabilities in no-treatment and post-treatment states (Table 3). The calibration process used data from 2013 to 2015.

**Calibration targets.** We selected three sets of calibration targets, which are very common across predictive models used in decision making, namely yearly counts of the total population with OUD, i.e., number of people with OUD in a given year including who also died that year, admissions to detox facilities, and fatal overdoses in MA from 2013-2015 (Table 1). These targets are related to different aspects of the model structure and are based on the quality of available data. Since total OUD counts cannot be observed directly from data, we used published estimates from a capture-recapture analysis [21]. Both detox admissions and fatal overdose count targets were directly observed from MA PHD. When calibrating to the yearly total OUD population, we used the 2015 total OUD counts and constrained our calibration to only select parameters that resulted in an increasing trend in OUD from 2013 to 2015. This is because the 2013 and 2014 total OUD counts from capture-recapture are not directly comparable to the 2015 estimates as they are based on slightly different data [1,21].

**Table 3. Calibration parameters.**

| Calibrated Parameters | Marginal Distribution | Source |
|---|---|---|
| Weekly non-stratified total arrival counts A(t) | | MA PHD* Range derived from the calibration process |
| $t = 2013$ | Uniform (6, 406)* | |
| $t = 2014$ | Uniform (7, 307)* | |
| $t = 2015$ | Uniform (356, 1356)* | |
| Detox transition rate multiplier $\eta(t)$ | | Arbitrary selected and the range derived from the calibration process |
| $t = 2013$ | Uniform (1, 2) | |
| $t = 2014$ | Uniform (1.25, 2.5) | |
| $t = 2015$ | Uniform (1, 2) | |
| Post-treatment initiation effects $\gamma_P$ | Uniform (0, 1) | Arbitrary selected range for probabilities |
| OUD state transition probabilities in no-treatment ($\rho_N$) and post-treatments ($\rho_P$) | Beta* | [24–27]* |
| Overdose rate multipliers | | |
| $m_N$ | Uniform (0.75, 1.75) | |
| $m_P$ | Uniform (1, 4) | |
| $m_T$ | | |
| $T = Buprenorphine$ | Beta* | [28] |
| $T = Naltrexone$ | | [28] |
| $T = Methadone$ | | [28,29]* |
| Weekly fatal overdose proportions **f(t)** | | MA PHD* |
| $t = 2013$ | Beta* | |
| $t = 2014$ | | |
| $t = 2015$ | | |

*See supplementary materials

Details of calibration targets are summarized in Table 1 and a detailed explanation of how these targets are derived is provided in the supplementary material.

**Calibration method.** We calibrated the model using an empirical approach. We used Latin Hypercube (LH) Sampling to efficiently search the multidimensional input parameter space. The algorithm accepts proposed parameter values with respective model outputs within pre-specified uncertainty ranges for each target. The range for the 2015 total OUD count target, is its published 95% confidence interval (CI) estimate [1,21]. The detox admissions and fatal overdose uncertainty ranges were defined to be within 10% of the observed count (Table 1). Additionally, we reject sampled parameters that do not result in an increasing trend in OUD counts from 2013 to 2015. Fig 2 depicts the flow diagram of the empirical calibration algorithm:

**Step 1:** Define plausible ranges and marginal distributions for each calibration parameter (Table 3). These ranges can be obtained from available data sources, based on domain expertise, or determined by the calibration (iterative) process (Appendix S2).

**Step 2:** Generate *N* sets of random values from the multidimensional parameter space using LH sampling and the marginal distributions from step 1.

**Step 3:** Run model simulations with proposed input parameter values from step 2 and compare model outputs to pre-specified calibration targets.

**Step 4:** Accept parameter values that resulted in model outputs within the uncertainty ranges of targets in the following order: 2015 total OUD → admissions to detox → Fatal

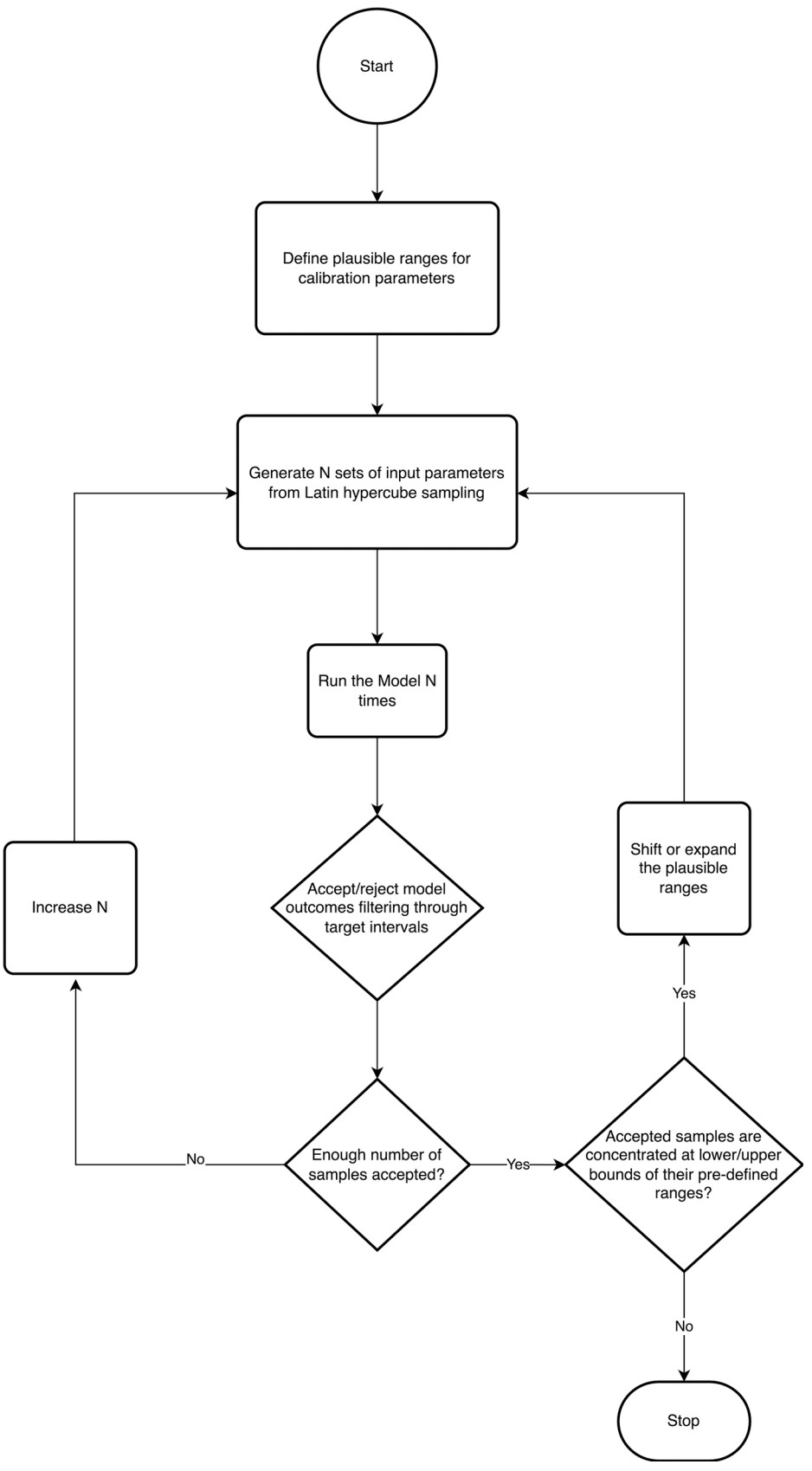

**Fig 2. Flow diagram of the empirical calibration algorithm.**

overdoses based on results from a sensitivity analysis indicating which RESPOND parameters have the biggest impact on simulated outcomes. Finally, accept parameters that result in an increasing trend in total OUD counts. Accepted parameter values need to pass all these checks.

**Step 5:** Increase $N$ and repeat step 2-5 until a sufficiently large sample of parameter values is collected, i.e., until no significant change is observed in the resulting distributions of calibration parameters and model outputs.

**Step 6:** Investigate the marginal distributions of accepted samples. If the values are concentrated at extremes of their pre-defined ranges, shift, or expand the ranges within plausible ranges and repeat steps 2-6.

**Step 7:** Stop the algorithm when full distributions are observed from accepted samples for marginal distributions of all calibration parameters.

One common problem that we observe when calibrating a complex predictive model to multiple calibration targets is that some model outputs never hit the respective targets. This can be an indication of a structural issue, e.g., the model fails to describe important components of the process.

Our empirical calibration results in sets of parameter values representing the empirical distribution of the calibrated parameters. To ensure that enough input samples were accepted from the calibration process, one needs to repeat the calibration process increasing $N$ to achieve a higher grid resolution of the multidimensional parameter space until no significant change is observed in the resulting distributions of the calibration parameters and the model outputs. If repeating the calibration process with increasing $N$ is computationally demanding, we suggest conducting a sensitivity analysis to define the size of a sufficiently large sample of parameters values for obtaining robust estimates of the outcomes of interest based on outputs from a well-calibrated model. This technique pertains to running the model for different sizes of random samples of the accepted parameter values until model outputs stabilize within the uncertainty ranges of the respective calibration targets.

## 2.4. Model validation

We validated the calibrated RESPOND model to evaluate the accuracy of model projections most important for the shared decision-making of OUD such as increasing access to MOUD. We used three validation approaches: internal, external, and face validity.

To validate externally, we compared model outcomes to data obtained directly from data sources or published studies that were not used in the model calibration process, including deaths due to competing risks, age-sex stratified total OUD and overdose counts, and total OUD counts for the entire simulation period (2013-2015). Our OUD population estimated from capture-recapture analysis accounts for unobserved OUD counts. Therefore, to obtain a comparable other deaths target for the external validation, we borrow findings from the literature as opposed to comparing directly to observed other death counts from MA PHD. Other studies have shown that deaths from other causes are usually 2.4-4.1 times the size of overdose deaths [30–32]. Therefore, we compared the other death counts calculated from the model to the other death counts obtained by applying the multipliers in the range 2.4-4.1 on fatal overdose count targets. We also validated model outcomes of 2015 age-sex stratified total OUD counts comparing those to estimates from the capture-recapture analysis [21]. Similarly, we validated model outcomes for stratified overdose counts (fatal and non-fatal combined) by comparing to the corresponding age-sex observed counts from MA PHD.

We assessed the face validity of model-generated year-end overdose counts (fatal and non-fatal combined) by comparing them to observed overdose counts from MA PHD during 2013-2015. Since non-fatal overdose counts are usually underreported, our external validation

aimed to ensure that the calibrated model produces overdose counts higher than the observed counts from MA PHD. The percentage of active users in different health states of the model was not available. However, according to clinical and epidemiological experts of OUD, about 90% of those in no-treatment should be active users. Similarly, the percentage of active users should be lower in treatments and higher in post-treatment episodes. Therefore, we assessed the face validity of active OUD percentages resulting from the RESPOND model.

The proposed empirical calibration approach also involves a heuristic iterative algorithm for identifying and potentially resolving issues arising from the external or face validation process. Starting with the entire set of calibration targets we adapt a backward approach for improving the results by excluding one target at a time starting from the last. After removing each target, we re-run the calibration, and given that validation targets were hit successfully, we investigate the discrepancies between the new accepted samples of parameter values with rejected samples from the previous implementation. Differences between the accepted and rejected samples in each step can be suggestive of changes required in model structure and/ or parameters so that the model can hit all calibration and validation targets. Structural issues with the model are in general harder to fixed as they may involve key changes in the under-lying distributional assumptions, simulation algorithm, etc. however they deemed necessary in cases where after an extensive calibration the model still fails to represent facts (observed outcomes) of the disease process for the entire population or specific subgroups.

## 3. Results

### 3.1. Model Calibration

We searched 5 million ($N$=5 million) input parameter sets (grid size: 5 million model runs $\times$ 102 model parameters), extracted from the multidimensional parameter space using LH sampling. Of these, 740 sets were accepted with the calibration algorithm (low acceptance rate). Model outcomes with all 5 million input parameters before ("*Pre-EC*") the calibration had a large variability as shown in Fig 3. As a result of the calibration process design, all model outcomes of 2015 total OUD counts, admissions to detox counts, and fatal overdoses resulted within their respective uncertainty ranges (S1 Table, Fig 3). Although slightly under-estimating the total OUD counts of 2013-14 estimated from the capture-recapture analysis, the model was able to produce results that closely match the respective total OUD count of 2015 (Fig 3(A)).

The red boxplots ("Target") present the actual distributions of the calibration targets based on point estimates and 95% CIs from the literature.

The grey boxplots ("Pre-EC") present the distributions of the calibration targets based on RESPOND simulations when using values for the model parameters randomly selected from the respective plausible ranges which determine the marginals of the multidimensional parameter space for implementing the LHS sampling design, prior to the Empirical Calibra-tion (EC) process. As expected, there is a large variability in the respective distributions of the simulated calibration targets.

The black boxplots ("Pre-EC") present the distributions of the calibration targets based on RESPOND simulations when using accepted values for the model parameters resulted from the Empirical Calibration ("EC") process.

We performed a sensitivity analysis to demonstrate that we generated enough LH samples to obtain accurate model outcomes. When we increase the number of samples in random sub-sets of accepted samples above 500 the empirical distribution of the model outcomes remains stable (S3 Fig), indicating that our final sample size of 740 is sufficient to capture the marginal distributions of the accepted parameter values.

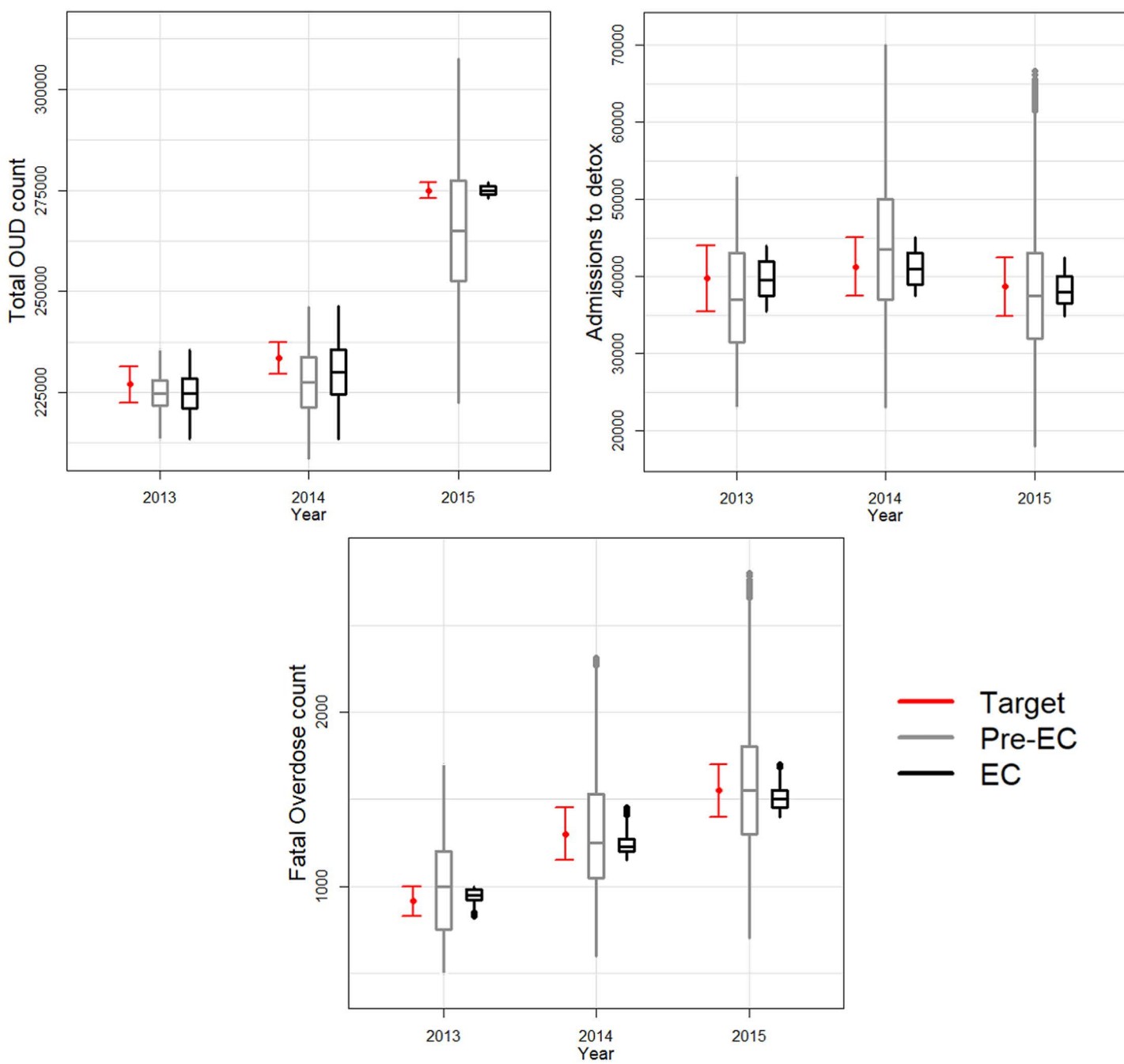

**Fig 3. Calibration results: Box-plots presenting the distributions of actual versus resulting (simulated) calibration targets ("Total OUD counts", "Admissions to Detox", "Fatal Overdose Counts") from each version of the calibrated RESPOND model over time.**

The calibration process resulted in an empirical joint distribution of the calibration parameters. The calibration process has substantially shifted the marginal distributions of several parameters with original non-informative uniform marginal distributions (Fig 4), including the 2014-15 total new OUD arrivals, detox rate transition multipliers, and no-treatment overdose rate multiplier. Prior to calibration, we derived informative marginal prior distributions

## Weekly non-stratified total arrival counts $A(t)$

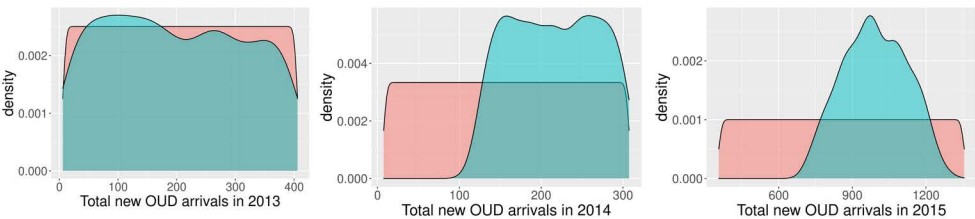

## Detox transition rate multiplier $\eta(t)$

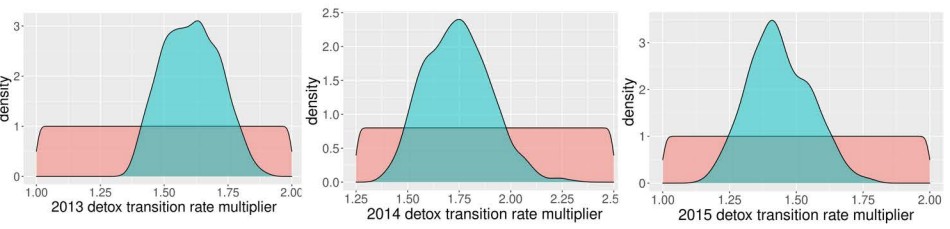

## Overdose rate multipliers $m_N$ and $m_P$

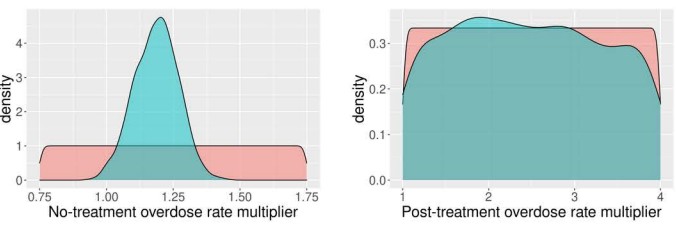

## Overdose rate multipliers $m_T$

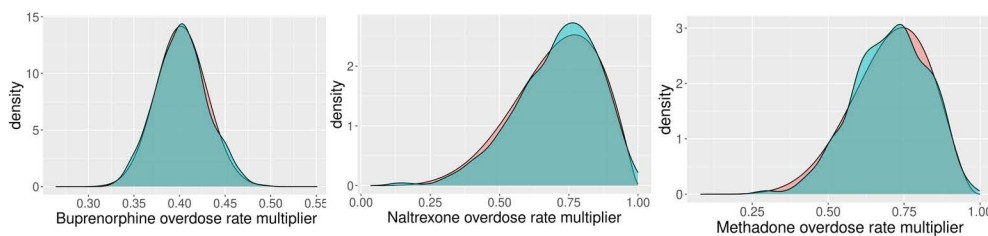

## Weekly fatal overdose proportions $f(t)$

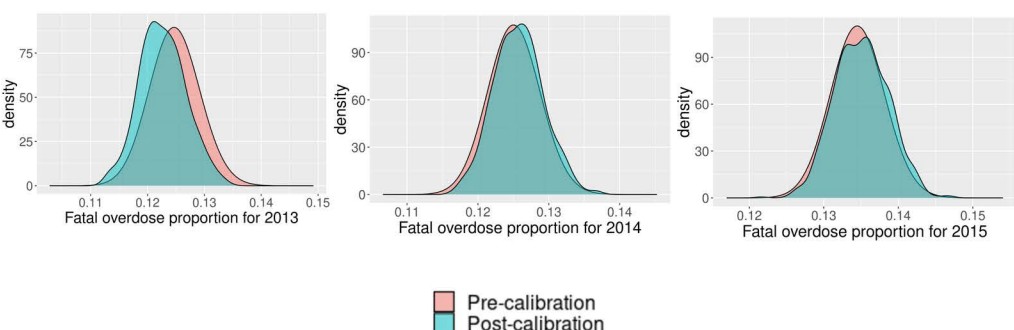

**Fig 4. Distribution of values for the calibrated parameters of the RESPOND model.** Results from the pre- and post- empirical calibration process.

from data for overdose rate multipliers in treatments, fatal overdose proportions, and OUD transition probabilities in no- and post-treatment. The calibration process did not change these starting distributions (Fig 4 and S4 Fig) implying that parameter ranges were already well-defined.

The blue density plots ("Pre-calibration") correspond to the marginal distributions specified by the plausible values for the calibrated RESPOND parameters which determine the multidimensional parameter space for implementing the LHS sampling design, prior to the Empirical Calibration (EC) process.

The pink density plots ("Post-calibration") correspond to the marginal distributions specified by the accepted values of the calibrated RESPOND parameters resulting in from the Empirical Calibration (EC) process.. Marginal distributions of remaining parameters (substance use transitions): $\gamma_P$, $\rho_N$ and $\rho_P$ are shown in Fig S4.

The calibration also failed to identify a more specific marginal distribution than the prespecified uniform distribution for 2013 total new OUD arrivals, post-treatment overdose rate multiplier, and initiation effects in post-treatment states. We did not incorporate direct estimates for the total OUD count in 2013 and 2014 in the calibration process likely leading to a dearth of information for new OUD arrivals of these two years although the last filtering step of selecting larger arrivals for 2014 compared to 2013. Similarly, we do not have health state-specific targets to calibrate post-treatment parameters. An overwhelming proportion of the OUD population was in no-treatment leaving only a small proportion in post-treatment (see S2 Table). Correspondingly, overdose counts from no-treatment were significantly higher compared to post-treatment due to a higher percentage of active users (see S3 Table). The model calculates yearly fatal overdose counts conditioned on total all overdoses (fatal and non-fatal combined), not on overdose counts specific to different health states. As a result, a higher proportion of fatal overdoses occur in no-treatment compared to post-treatment. Since fatal overdose counts are a target, the calibration was dominated by the fatal overdoses of no-treatment leading to an accurate calibration of the no-treatment overdose rate multiplier and uninformative calibration of the post-treatment overdose rate multiplier.

When looking at the correlation coefficient between the accepted values of the calibrated parameters we found a strong negative correlation (r = -0.89) between new OUD arrival counts of 2013 and 2015, and much weaker correlations between 2014 and 2015, and between 2013 and 2014 (r = -0.41 and r = 0.005, respectively).

### 3.2. Model Validation

With 740 accepted values for the calibrated model parameters, we ran RESPOND and computed year-end overdose counts for 2013-15. Fig 5 shows the distribution of all overdose model outcomes compared to observed counts from MA PHD. Since overdose counts are usually underreported, we expect the model to result in higher overdose counts compared to what is observed. However, as shown in Fig 5, the model only overestimated the overdose counts in 2013, and for both 2014-15 the model underestimated the overdose counts on average.

To investigate this unexpected result, we excluded all the fatal overdose targets from the calibration process and found that the model produced more variable overdose counts that aligned more closely with the observed counts (see S5(A) Fig). Subsequently adding the 2013 fatal overdose count target led to the underestimation of the 2014 and 2015 overdose counts (see S5(B) Fig). Just after adding the 2013 fatal overdose target in the calibration, we found that accepted samples of no-treatment overdose multiplier had non-overlapping ranges to rejected samples (see S5(C) Fig). Our model uses only one multiplier for no-treatment overdose rates from 2013-15. However, accepted samples of this non-time varying multiplier can hit only the 2013 overdose counts well, and rejected samples correspond to the model

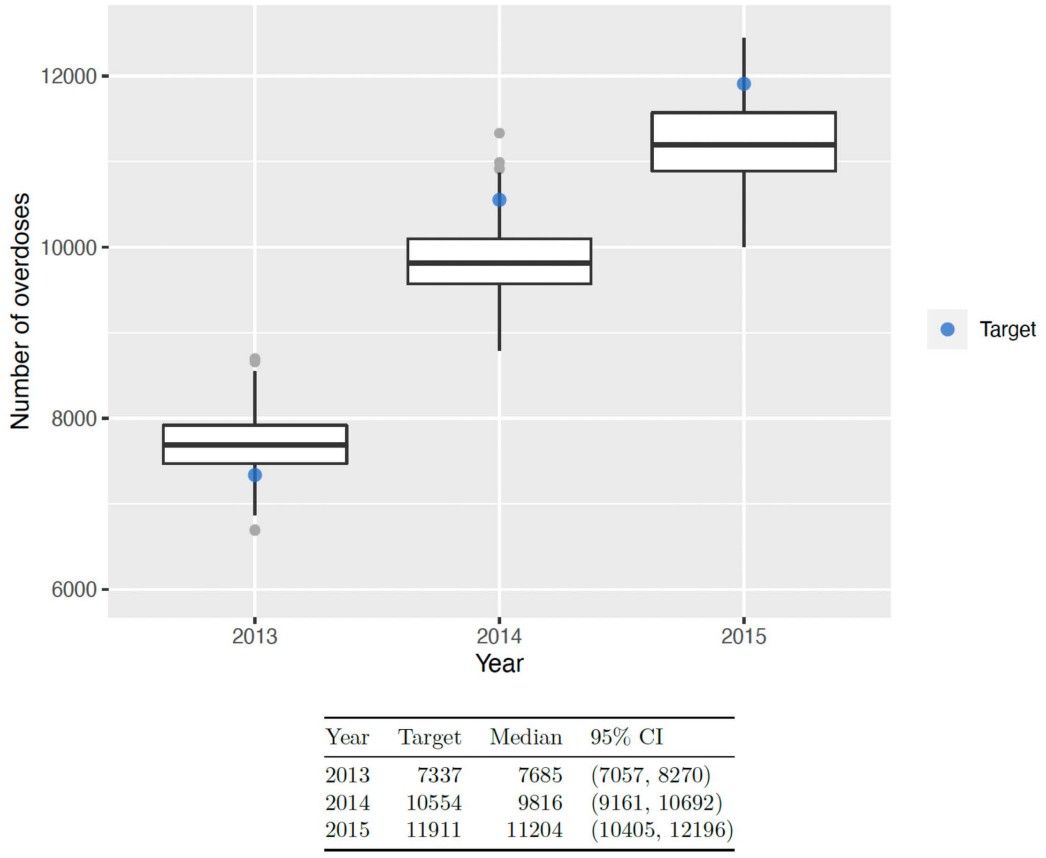

| Year | Target | Median | 95% CI |
|------|--------|--------|--------------|
| 2013 | 7337 | 7685 | (7057, 8270) |
| 2014 | 10554 | 9816 | (9161, 10692) |
| 2015 | 11911 | 11204 | (10405, 12196) |

**Fig 5. Comparison of year-end all overdose counts (non-fatal and fatal combined) model outcomes to observed overdose count targets from MA PHD.**

outcomes that could hit the counts of 2014-15 well but not 2013. This finding suggests that we require time-varying multipliers to produce higher overdose counts for 2014 and 2015.

To validate other cause mortality rates, we obtained data from studies reporting overdose death rates and multiplied these by 2.4-4.1 to obtain estimated external target ranges for other deaths due to competing risks [30–32]. Model predictions of 2014 and 2015 other cause mortality lie within the respective calibration target ranges (Fig 6).

The calibrated model predicted a median percentage of people who were actively using opioids of 72% (CI: [68%, 77%]) by the end of the year 2013. This declined to 66% (CI: [61%, 75%]) in both 2014 and 2015 (Fig 7(A)). The median active percentage in no treatment declined from 81% in 2013 to 72% in 2015 but remained stable at 27% and 74% for all treatments and post-treatment episodes respectively (Fig 7(B), S4 Table). Although the model estimated an active percentage in no-treatment lower than expert opinion (90%), it predicted a higher (always above 64% from all accepted samples) active OUD percentage as expected (Fig 7(B)). Additionally, the model projects a lower percentage of active OUDs in treatments and a higher percentage in post-treatments meeting face validity criteria.

## 4. Discussion

In this study we present in detail an example of calibrating and validating a dynamic, state-transition, cohort-based model that simulates the OUD population in Massachusetts. We

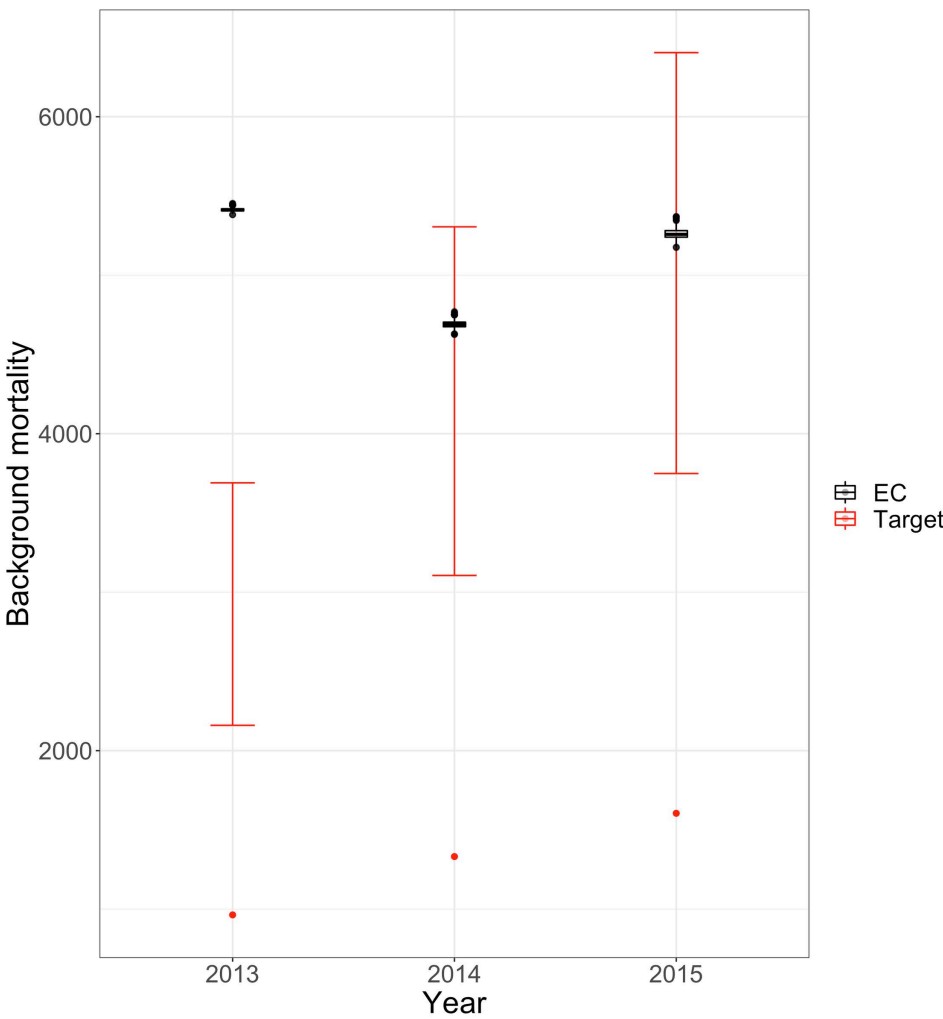

**Fig 6. Other cause mortality model outcomes in comparison to target death counts that are corrected to be between 2.4-4.1 times the size of overdose deaths.** Label EC represents death counts due to competing risks resulted from the empirically calibrated model. Red dots and red error bars represent the observed counts from MA PHD and the corrected ranges respectively.

utilized an empirical calibration approach to fit the model to multiple targets, using LH sampling to search within the multidimensional input parameter space. The proposed algorithm results in sets of accepted values for each calibrated parameter thus allowing for quantifying and conveying the parameter uncertainty to the model outputs. From this empirical calibration, we discovered a set of non-identifiable parameters including post-treatment parameters due to the model's structural constraints and lack of available calibration data. The calibrated model provided a good fit for all pre-specified calibration and validation targets. The fit was particularly good for total OUD counts, fatal overdose counts, and other cause mortality in 2015. Therefore, the model outcome of alive OUD counts of 2015 is an accurate prediction of actual alive OUDs in 2015 which can be calculated by subtracting fatal overdose count and deaths due to competing risks from the total OUD count.

Calibrating complex population models with many parameters is challenging especially with limited target data. In this paper, we have proposed an empirical approach to calibrating

**(A) Overall Active Percentage**

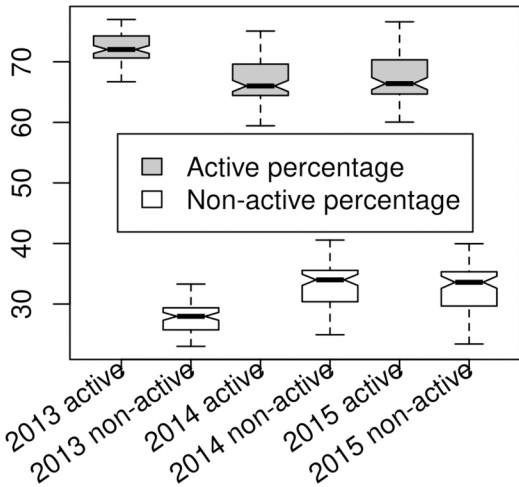

**(B) Percentage of Active vs Non-active OUD Counts in Different Health States**

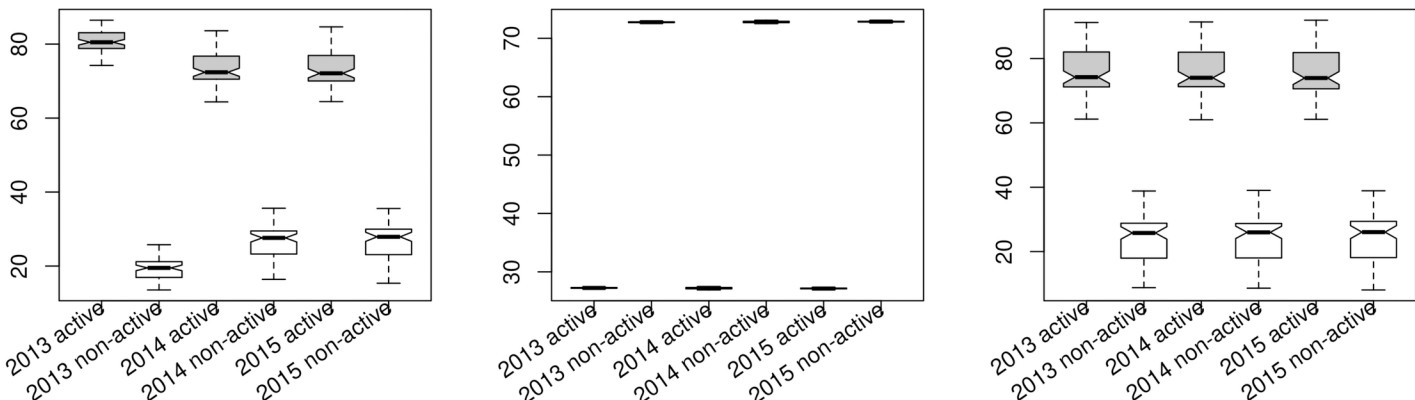

**Fig 7. Active vs non-active OUD percentages resulted from the calibrated model.** (A) Overall active vs non-active percentages across all health states each year. (B) Yearly percentages of active vs non-active users in different health states.

complex simulation models. The flexibility of the proposed algorithm allows us to better understand the model structure, identify structural issues, and explore underlying relationships between the model parameters.

The proposed algorithm has some limitations. The algorithm can only accept parameter values within the pre-specified range of a marginal distribution and cannot shift the distribution outside of its initial range to fit calibration targets. Therefore, if we represent available information for a model parameter using a marginal distribution with a very narrow plausible range, we may not hit calibration targets. In this case, we suggest expanding the range of the prior and re-running the algorithm. This is distinct from a Bayesian approach, or a mixed type, directed search optimization method combined with LH sampling which can shift parameter ranges within a single calibration run by design. However, with such methods, it is usually difficult to achieve convergence for a complex model like RESPOND. The proposed empirical approach can reveal structural issues of the model and the resulting set of accepted

parameter values determine a multidimensional space from which randomly selected vectors have higher chances of producing model simulations close to the pre-specified calibration targets. This "focused" multidimensional space can inform the priors of a more comprehensive Bayesian calibration method. The proposed empirical approach is a grid-based approach that requires generating a huge amount of LH samples to capture plausible values of input parameter space, as the current filtering approach accepts samples within the uncertainty intervals of each calibration target sequentially. This becomes particularly challenging when extending the calibration period by incorporating targets for more years or when calibrating to age-sex stratified targets. Filtering through target ranges sequentially decreases the number of accepted input samples at each filtering step and with many targets we may end up with very few samples or no samples accepted from the calibration process, forcing us to increase the size of the LH grid. Furthermore, accepting inputs based on target ranges is also subject to overfitting. An approach that fits model outcomes to the trends in target data can help us to avoid both overfitting and computational burden.

The calibration exercise presented here revealed a set of non-identifiable parameters, particularly in post-treatment (Fig 4 and S4 Fig; parameters with flat distributions of accepted values). This could be due to lack of available information such as post-treatment specific target data, as well as issues related to the model structure. The resulting samples of these non-identifiable parameters are still valid as they produce accurate predictions for the model outcomes of interest. However, the model is not well calibrated for applications that primarily focus on post-treatment-specific model outcomes. Further, for no-treatment and post-treatment, we did not have good information on prior OUD transition probabilities of different age groups and our calibration failed to provide age-specific parameter ranges for these probabilities (Fig S4). One implication of this is that we may not need as many age categories since there is not sufficient data to parameterize and calibrate.

One of the limitations in the RESPOND model parametrization is that we assume constant time variability throughout a year for certain parameters such as new OUD arrivals, and overdose rates. This framework allows us to carry out a cost-effective analysis comparing different strategies by projecting future outcomes with the assumption that epidemiological trends observed at the end of the calibration period continue throughout the rest of the simulation. However, it prevents us from making realistic forecasts such as model projection of OUD population size after 2015. Utilizing data for more years, the parametrization of the RESPOND model can be improved by incorporating approaches for longitudinal data to model time-dependent parameters.

The opioid epidemic and overdose crisis are constantly evolving, and it is impossible to predict future developments. For example, the introduction of synthetic opioids, like fentanyl, drove a major upward inflection in overdose rates around the U.S. that could not have been predicted by any simulation model prior to that event occurring. The calibrated RESPOND model can be used for projections outside the prediction period. However, as in any predictive model, projections are subject to additional uncertainty and based on the strong assumption that the trends fitted with the calibrated model remain unchanged outside the calibration data and period. Model projections should always be used with caution, especially within an area where things are rapidly changing such as the OUD epidemic (changes in laws on drug use and possession, increasing use of fentanyl, changes in MOUD, etc.).

Similarly, future changes in the supply or substances, or substance use behaviors and preferences could impact the accuracy of projections. Similarly, the COVID-19 pandemic likely interrupted OUD treatment and drug use patterns and resulted in discontinuity of calibrated trends. As data become available, however, this method makes it possible to rapidly incorporate COVID-era trends into the simulation.

In the current analysis, even though RESPOND models four types of treatments for OUD, we have only incorporated admission to detox as a calibration target. This is because we did not have access to MA PHD to obtain admission counts of other MOUDs at the time of this analysis. Once available, admission to MOUD counts can be used as external targets to validate admission counts resulting from the calibrated model (S6 Fig). Incorporating these admission counts as additional targets in our future calibration work should help us to calibrate treatment transition rates well.

One of the strengths of the RESPOND model is its ability to produce age-sex stratified model outcomes. Depending on the use case of the model, we may require calibrating the model to stratified targets. Current calibration did not consider stratified target counts and as a result, the model didn't produce matching counts for certain outcomes (see S1 and S2 Fig). In a more rigorous Bayesian calibration, we plan to incorporate age-sex stratified targets to produce accurate model outcomes.

## Supporting Information

**S1 File. RESPOND: Empirical Calibration Supplementary material.**
(DOCX)

**S2 File. RESPOND Technical Appendix.**
(DOCX)

**S3 File. List of Figs.**
(DOCX)

## Acknowledgments

The authors would like to thank Alexandra Savinkina for helping us to find data for model validation and Emily Stewart for doing a comprehensive copy edit of the manuscript. We also thank Matthew Carroll and Dimitri Baptiste for their insightful comments and suggestions during the revision process. We acknowledge the Massachusetts Department of Public Health for creating the unique, cross-sector database used for this project and for providing technical support for the analysis.

## Author contributions

**Conceptualization:** R. W. M. A. Madushani, Benjamin P. Linas, Laura F. White, Stavroula Chrysanthopoulou.

**Data curation:** R. W. M. A. Madushani, Jianing Wang.

**Formal analysis:** R. W. M. A. Madushani, Jianing Wang, Michelle Weitz.

**Funding acquisition:** Benjamin P. Linas, Laura F. White.

**Investigation:** R. W. M. A. Madushani, Jianing Wang, Laura F. White, Stavroula Chrysanthopoulou.

**Methodology:** R. W. M. A. Madushani, Jianing Wang, Benjamin P. Linas, Laura F. White, Stavroula Chrysanthopoulou.

**Project administration:** Benjamin P. Linas.

**Resources:** Benjamin P. Linas.

**Software:** R. W. M. A. Madushani, Jianing Wang.

**Supervision:** Benjamin P. Linas, Laura F. White, Stavroula Chrysanthopoulou.

**Validation:** R. W. M. A. Madushani, Benjamin P. Linas, Laura F. White, Stavroula Chrysanthopoulou.

**Visualization:** R. W. M. A. Madushani, Jianing Wang, Michelle Weitz.

**Writing – original draft:** R. W. M. A. Madushani, Stavroula Chrysanthopoulou.

**Writing – review & editing:** R. W. M. A. Madushani, Jianing Wang, Michelle Weitz, Benjamin P. Linas, Laura F. White, Stavroula Chrysanthopoulou.

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
