## [Decision Letter · Decision Letter 0]

18 Jun 2024

PONE-D-24-12466Empirical calibration of a simulation model of opioid use disorderPLOS ONE

Dear Dr. Chrysanthopoulou,

Thank you for submitting your manuscript to PLOS ONE. After careful consideration, we feel that it has merit but does not fully meet PLOS ONE’s publication criteria as it currently stands. Therefore, we invite you to submit a revised version of the manuscript that addresses the points raised during the review process. **I believe that the reviewers converged on similar questions regarding calibration as well as the need to justify some of the assumptions, and parameters. I also agree with Reviewer #1 regarding revising the figures.** Please submit your revised manuscript by Aug 02 2024 11:59PM. If you will need more time than this to complete your revisions, please reply to this message or contact the journal office at plosone@plos.org . Please include the following items when submitting your revised manuscript:

We look forward to receiving your revised manuscript.

Kind regards,

Kimberly Page, PhD, MPH

Academic Editor

PLOS ONE

Journal Requirements:

This work was funded by the National Institute on Drug Abuse (R01DA046527)

(P30040500) and the National Institute of General Medical Sciences (R35GM141821).

Reviewers' comments:

Reviewer's Responses to Questions

**Comments to the Author**

1. Is the manuscript technically sound, and do the data support the conclusions?

Reviewer #1: Yes

Reviewer #2: Yes

2. Has the statistical analysis been performed appropriately and rigorously? 

Reviewer #1: N/A

Reviewer #2: I Don't Know

3. Have the authors made all data underlying the findings in their manuscript fully available?

Reviewer #1: Yes

Reviewer #2: Yes

4. Is the manuscript presented in an intelligible fashion and written in standard English?

Reviewer #1: Yes

Reviewer #2: Yes

5. Review Comments to the Author

**Reviewer #1: ** Manuscript Number: PONE-D-24-12466

Title: Empirical calibration of a simulation model of opioid use disorder

In this article a dynamic model that simulates OUD population in Massachusetts is developed. The model is calibrated using 3 targets related to OUD data. The model includes different health and treatment statutes. One of the aims of the outcome of this study is to provide valuable information to health institutions.

The topic of the article is very important and worthy of study. The calibration of the model to target data is complex since there are many uncertain parameters in the model. The authors developed a plausible method to estimate the parameters. Although as the authors mentioned there are some non-identifiable parameters that can affect the outcomes of this study. The authors mentioned some limitations of this study which is important.

The article can be improved in many ways so readers can grasp the main ideas of the paper without reading it several times.

These are some of the aspects that should be addressed.

The presentation of Figure 1 should be improved. Adding the parameters to the diagram would help the readers to understand the model in an easier way. For instance, the Lambdas.

Maybe I missed but explain why there is no transitions between nonactive no-injection and nonactive injection.

The 8 different transition probabilities in the core simulation (a, b, c, d, e, f, g and h) should be included in one of the Figures so readers can grasp better the model and dynamics.

The presentation of Figure 3 should be improved. It is difficult to differentiate between target, EC and Pre-EC.

Please clarify about the two sources of mortality in RESPOND model. Please make it clear that there is no overlap between these two types of deaths. Does the real data provide nonoverlap mortality data?

A better explanation can be given to the cohort initiation. Especially the last sentence

“The balance of arrivals and deaths results in the total Massachusetts OUD population size”

The model is very difficult to calibrate since there are some functions that need to be calibrated.

Section 2.1. It is not clear the type of model. Please add a paragraph about the type of model. The process shown in

Figure 1 can be achieved with different type of models.

Figure 1. It is not clear why the transition to the death state occurs at the same rate beta for individuals in no-treatment, treatment and post-treatment. It seems that should be different. This requires a good explanation in the manuscript.

Table 1. Please add more details about how the year OUD count is done. Please show the values of the 3 sources of OUD count. No overlap between them?

Table 3. Not sure why for function f(t) it shows beta? Is this a typo?

The Tables can show the units of the parameters.

A brief explanation of the capture-recapture analysis to estimate total OUD counts should be included in order to have some self-content.

Calibration method. Please explain why there is a specific order? If the model’s output does not satisfy any of the 3 target ranges, then the parameter set is discarded?

Page 22. Ther authors mentioned that the algorithm can suggest changes un the structure of the model which is much harder than changing ranges for the parameters. Please explain this better and if there was a structural change when the calibration was implemented.

Some more specific analysis/comments about the obtained distributions of the parameters should be added.

In some way the calibration method uses a Bayesian approach. Please comment on this in the conclusions. For instance, what would happen if a strict Bayesian approach were used with priors similar to the ones obtained in this study? Or with very wide ranges for the priors.

In the conclusions the authors mentioned that a set of non-identifiable parameters was found. Please mention this particular set to have a better global understanding of the results of this study.

**Reviewer #2: ** This is a very interesting manuscript discussing the calibration of an opioid use disorder model. It is important as it is an area in which data can be limited and modeling can fill a very important need. Calibration can provide more accurate modeling forecasts which are important for decision makers.

While I think the analysis is important, I find that there are some major limitations to this calibration which need to be addressed (or at least detailed within the limitations and explained).

Introduction:

The focus of the introduction seems to be the importance of modeling and the public health emergency that is opioid use disorder. I suggest focusing more on the importance specifically of calibrating models, and the difference between manual and empiric calibration.

Paragraph 2 needs citations.

Methods:

The major weakness within this analysis is that the model was only calibrated to 3 years of data, and that data comes from 2013-2015. Limiting data to only 3 years can make it difficult to see trends and calibrate to true trends, and may be the reason that the calibration struggled to meet targets. Why was the decision made to stop here? I assume it is a data limitation. Regardless, this needs to be explained within the manuscript methods AND listed within the limitations.

Similarly, there needs to be a defense of using a model calibrated to data from a decade ago to forecast the future, especially within an area where things are rapidly changing (ie laws on drug use and possession, increasing use of fentanyl, changes in MOUD).

Is relapse estimated to be the same no matter how long someone has been in inactive drug use (ie is my risk of relapsing the same when I last used opioids 1 month ago vs 5 years ago)? Either way, this needs to be explained. The same question with treatment end- am I as likely to leave treatment after 1 month as after 5 years?

Why were the calibration targets chosen? There is some brief discussion of this, but not specific reasoning for each of the three targets. Why was OUD number chosen when you state it is unobservable AND estimates for 2013-2014 aren’t comparable to 2015? Why were detox admissions chosen?

Please further describe methods for internal validation. If this is in the supplement, point to it in the manuscript.

“To validate externally, we compared model outcomes to data obtained directly from data sources or published studies that were not used in the model calibration process, including deaths due to competing risks, age-sex stratified total OUD and overdose counts, and total OUD counts for the entire simulation period (2013-2015).” Isn’t total OUD counts for the simulation period a calibration target? How can it also be an external validation target?

Discussion:

Limitations need to be expanded to incorporate some of the issues mentioned above.

6. PLOS authors have the option to publish the peer review history of their article (what does this mean? ). If published, this will include your full peer review and any attached files.

**Do you want your identity to be public for this peer review?** For information about this choice, including consent withdrawal, please see our Privacy Policy .

Reviewer #1: No

Reviewer #2: No

---

## [Author Response · Author response to Decision Letter 0]

3 Sep 2024

We have provided detailed responses to all reviewers' comments, which can be found in the submitted "Response to Reviewers" letter attached with this 2nd submission.

---

## [Editor Report · Decision Letter 1]

6 Sep 2024

Empirical calibration of a simulation model of opioid use disorder

PONE-D-24-12466R1

Dear Dr. Chrysanthopoulou%,

We’re pleased to inform you that your manuscript has been judged scientifically suitable for publication and will be formally accepted for publication once it meets all outstanding technical requirements.

Kind regards,

Kimberly Page, PhD, MPH

Academic Editor

PLOS ONE
---

## [Editor Report · Acceptance letter]

PONE-D-24-12466R1

PLOS ONE

Dear Dr. Chrysanthopoulou,

I'm pleased to inform you that your manuscript has been deemed suitable for publication in PLOS ONE. Congratulations! Your manuscript is now being handed over to our production team.

Kind regards,

on behalf of

Dr. Kimberly Page

Academic Editor

PLOS ONE